# Utilizing Mixed Training and Multi-Head Attention to Address Data Shift in AI-Based Electromagnetic Solvers for Nano-Structured Metamaterials

**DOI:** 10.3390/nano13202778

**Published:** 2023-10-17

**Authors:** Zhenjia Zeng, Lei Wang, Yiran Wu, Zhipeng Hu, Julian Evans, Xinhua Zhu, Gaoao Ye, Sailing He

**Affiliations:** 1National Engineering Research Center for Optical Instruments, Centre for Optical and Electromagnetic Research, Zhejiang University, Hangzhou 310058, China; 22030008@zju.edu.cn (Z.Z.); le1wang@zju.edu.cn (L.W.); 22130066@zju.edu.cn (Y.W.); miniwho@zju.edu.cn (Z.H.); julian.evans@colorado.edu (J.E.); 2Shanghai Institute for Advanced Study, Zhejiang University, Shanghai 201203, China; zhuxh@zju.edu.cn; 3Taizhou Research Institute, Zhejiang University, Taizhou 317700, China; yegaoao@126.com; 4Department of Electrical Engineering, Royal Institute of Technology, 100 44 Stockholm, Sweden

**Keywords:** data shift, mixed training, multi-head attention, AI-based electromagnetic solvers, nano-structured metamaterials

## Abstract

When designing nano-structured metamaterials with an iterative optimization method, a fast deep learning solver is desirable to replace a time-consuming numerical solver, and the related issue of data shift is a subtle yet easily overlooked challenge. In this work, we explore the data shift challenge in an AI-based electromagnetic solver and present innovative solutions. Using a one-dimensional grating coupler as a case study, we demonstrate the presence of data shift through the probability density method and principal component analysis, and show the degradation of neural network performance through experiments dealing with data affected by data shift. We propose three effective strategies to mitigate the effects of data shift: mixed training, adding multi-head attention, and a comprehensive approach that combines both. The experimental results validate the efficacy of these approaches in addressing data shift. Specifically, the combination of mixed training and multi-head attention significantly reduces the mean absolute error, by approximately 36%, when applied to data affected by data shift. Our work provides crucial insights and guidance for AI-based electromagnetic solvers in the optimal design of nano-structured metamaterials.

## 1. Introduction

Nano-structured metamaterials [1,2] exhibit exceptional light field manipulation capabilities and optical properties such as negative refraction [3], perfect absorption [4], and surpassing the diffraction limit [5] through carefully designed configurations. Currently, nano-structured metamaterials find extensive application in fields such as optical communications [6], super-resolution imaging [7], and optical computing [8]. The key to realizing novel functionalities in nano-structured metamaterials lies in the precise design of their structural configurations [9]. Traditional optimization design approaches often involve parameter scanning or optimization algorithms to iteratively optimize the device structure for desired functionalities [10,11]. These approaches typically employ complex numerical simulation techniques, such as the finite-difference time-domain (FDTD) method [12], the finite element method [13], and rigorous coupled-wave analysis [14], etc., to compute the spectral response and optical properties of the constructed photonic structures. These numerical methods necessitate fine meshing and iterative updating of the nano-structures, which requires a complex modeling process and substantial computational resources to obtain corresponding electromagnetic response results [12,13,14]. In large-scale parameter space exploration, conventional optimization design methods often require hundreds or thousands of numerical simulation to attain the desired nano-structure [15]. Each numerical simulation typically takes several hours or days, and as the complexity of the metamaterial increases, the required time grows exponentially [15,16].

The adoption of deep learning models as a substitute for numerical simulators is an effective approach for achieving the rapid and efficient design of nano-structured metamaterials [17]. Deep learning involves the sequential stacking of multiple sets of nonlinear layers and, once trained, deep learning models have the theoretical capability to fit any nonlinear relationships or hidden rules [18,19]. In deep learning, matrix operations are employed between layers, allowing for the rapid and efficient calculation of the optical properties of nano-structured metamaterials [17]. The solving speed can be improved by 2 to 5 orders of magnitude compared to traditional numerical simulators [17,20]. Presently, deep learning has widely replaced numerical simulators in the rapid and efficient design of various functional metamaterials, including thermal emitters [21], grating couplers [22], plasmonic nano-structures [23], power splitters [24], and chiral metamaterials [25], etc.

In studies that employ deep learning in lieu of numerical simulators, which are called AI-based electromagnetic solvers [26], a subtle phenomenon arises where deep learning is often trained on a training set generated using a random process for nano-structures, but it is ultimately applied to predict selected/designed nano-structures in the optimization process [21,22,23,24,25]. For instance, in the design of a grating coupler, randomly generated grating couplers generally exhibit lower coupling efficiency, while the efficiency of grating couplers selected through optimization is significantly higher than those that are randomly generated. This phenomenon bears resemblance to the concept of data shift in the field of machine learning, and we refer to it here as ’data shift’ [27]. To the best of our knowledge, this phenomenon has not been explored in the realm of photonic metamaterials or device design.

In this work, we investigate the presence, impact, and solutions for data shift in AI-based electromagnetic solvers for nano-structured metamaterials. Taking a one-dimensional grating coupler as an example, we compare the coupling efficiency distributions of randomly generated device structures with those selected through optimization algorithms. Our analysis reveals substantial discrepancies in the coupling efficiency distributions and employs kernel principal component analysis (KPCA) to understand their structural differences. We demonstrate that data shift can significantly diminish the predictive performance of deep learning models for electromagnetic solvers. To address the issue, we propose three effective strategies to mitigate the effects of data shift, namely, mixed training, adding multi-head attention, and a comprehensive approach that combines both strategies. Our experimental evaluation, based on multiple metrics including mean absolute error (MAE), mean squared error (MSE), and R-squared (R${}^{2}$), validates the effectiveness of the proposed methods.

## 2. Metamaterial Structure and Datasets

To illustrate the presence of data shift in AI-based electromagnetic solvers, we selected a one-dimensional non-uniform grating coupler as an example for nano-structured metamaterials, employing the classical SOI (silicon on insulator) structure as the fundamental configuration for the grating coupler. The schematic diagram of the grating coupler is depicted in Figure 1. The top-layer grating features a full-etch structure with a thickness of 220 nm and a length of 12 μm. For ease of fabrication, the grating structure is uniformly divided into 120 individual units, each with a size of 100 nm, with material assigned as either air or Si for each unit. The buried oxide layer material is SiO${}_{2}$, with a thickness of 2 μm. A Gaussian light beam in TE mode, in which the electric field component is entirely transverse (perpendicular) to the direction of wave travel, is incident from an SMF-28 optical fiber [28] into the center of the grating, operating at a wavelength of 1550 nm with a mode field diameter of 10.4 μm. To minimize the second-order Bragg reflection of the grating coupler, we set the incident angle of the Gaussian light to 5${}^{\circ }$. We employ Lumerical FDTD to calculate the coupling efficiency of the grating couplers at 1550 nm. The mesh refinement method we use is “conformal variant 0”, which can enhance simulation speed and accuracy for a given mesh and is used for most FDTD simulations except interfaces involving metals or PEC material [29]. We set all the boundary conditions to “PML” (perfectly matched layer), which aims to emulate the characteristics of electromagnetic wave propagation in homogeneous space and can effectively absorb electromagnetic waves at various incident angles, ensuring the accuracy of the numerical simulations [30,31].

We consider a scenario where a deep learning model is used as an electromagnetic solver to help in designing a grating coupler operating at a wavelength of 1550 nm. The randomly generated grating structures along with their corresponding coupling efficiencies (the FOM, figure of merit, which is a metric to determine which structure is optimal or most suitable for a specific problem) are used as datasets for training and testing the deep learning model. The trained deep learning model is then used to rapidly give the coupling efficiency of each new grating structure generated using an optimization method (e.g., the genetic algorithm [32]). To generate the required training and testing datasets, the entire grating structure is divided into 120 uniform units and represented by a one-dimensional binary vector of length 120, where ‘0’ represents air and ‘1’ represents silicon. The entire dataset consists of two parts: one is a binary vector of length 120, representing the grating structure as input, and the other is the coupling efficiency at 1550 nm as the label. We generated a total of 200,000 sets of random data and split them in an 8:2 ratio, resulting in 160,000 sets for training and 40,000 sets for testing. Since the test set in this part is similar to the training set, we call it a similar test set (test${}_{s}$ set). The generation process of the training set and similar test set is depicted in Figure 2a. To illustrate the presence and impact of data shift during the optimization process, we employed a genetic algorithm with coupling efficiency at 1550 nm as the FOM, to iteratively optimize the grating coupler’s structure. In this process, we saved and selected grating structures along with their corresponding coupling efficiencies generated during the iterative optimization process. We selected the top 10,000 efficient structures to simulate the device structures (input) and their corresponding coupling efficiencies (labels) that the deep learning model needs to predict during optimization using the genetic algorithm. Since the coupling efficiency of this portion of the data is higher than that of randomly generated data, we refer to it as the high-efficiency test set (test${}_{h}$ set), as illustrated in Figure 2b. The genetic algorithm we used is implemented using the Python package scikit-opt, while data transfer between Python and Lumerical FDTD was accomplished through the latter’s built-in Python API.

## 3. Data Shift

In the field of deep learning, data shift refers to the phenomenon where the distribution of input data changes between the training and testing phases of a deep learning model, leading to suboptimal performance of the model on new input data [27]. This variation in data distribution can be caused by factors such as different time periods [33], varying experimental conditions [34], or diverse data sources [35]. Data shift can have a detrimental impact on the performance of deep learning models because the features and patterns learned during the training phase may no longer be applicable during the testing phase, thereby reducing the model’s generalization ability [27]. Currently, in the field of deep learning, methods for mitigating data shift include domain adaptation [36], transfer learning [35], and adversarial training [37].

For AI-based electromagnetic solvers for nano-structured metamaterials, we observe a similar phenomenon of data shift. A typical scenario involves training a deep learning model on a training set composed of randomly generated structures and their corresponding optical responses, and then applying it for predictions on selectively optimal structures during the optimization process. We illustrate the data shift issue using the training set, the similar test set, and the high-efficiency test set previously generated. Figure 3a,b depict the differences in label (coupling efficiency) distributions among the training set, the similar test set, and the high-efficiency test set. The probability density plots of the training set and similar test set are nearly identical, with labels primarily concentrated between 0 and 0.05, and an average coupling efficiency of 0.76%. There is a notable disparity in the probability distribution plots between the training set and the high-efficiency test set. The labels in the high-efficiency test set are predominantly distributed between 0.09 and 0.2, with an average coupling efficiency of 12.77%. This indicates a clear data shift between the high-efficiency test set and the training set in terms of labels. To further demonstrate the data shift in terms of inputs (grating structures) among the training set, the similar test set, and the high-efficiency test set, we randomly selected 1000 samples from each set. We employ KPCA [38] to compress their input vectors of length 120 into two components, and measure their average reconstruction error using MSE. The MSE for the training set is 0.49, for the similar test set is 0.49, and for the high-efficiency test set is 0.43. This indicates that the two principal components after compression effectively retain their original information. Figure 3c,d display the distributions of these principal components. The distributions of principal components on the training set and the similar test set are nearly identical. However, there is a significant disparity in the distribution of principal components between the training set and the high-efficiency test set. The principal components in the training set are more concentrated, while those in the high-efficiency test set are more dispersed. This signifies a clear data shift in terms of inputs between the training set and the high-efficiency test set.

## 4. Deep Learning Model Architecture

The data shift issue in AI-based electromagnetic solvers possesses distinct characteristics from other types of data shift in machine learning. The optimized and filtered devices, along with their corresponding optical characteristics, conform to the same underlying principles—Maxwell’s equations. This is a unique feature not commonly found in other instances of data shift in the machine learning domain. Additionally, data shift in AI-based electromagnetic solvers is relatively predictable. For instance, when optimizing grating couplers at a working wavelength of 1550 nm, one can expect that efficient devices will exhibit data shift.

Both the optimized and randomly generated devices, along with their corresponding optical characteristics, conform to Maxwell’s equations. Consequently, the effective alleviation of data shift in AI-based electromagnetic solvers can be achieved if deep learning models can better grasp the representations of Maxwell’s equations. The deep learning models should possess strong data fitting capabilities and robust global learning abilities. Therefore, we adopted ResNet [39] as the foundational framework for our model and integrated a multi-head attention [40] to enhance the neural network’s global learning capability. ResNet, introduced by Kaiming He et al. [39,41] in 2015, is a neural network architecture. By employing residual connections, ResNet effectively mitigates the challenges of vanishing gradients and exploding gradients, enabling the construction of deep neural networks [39]. ResNet modularizes the residual connections, simplifying the process of increasing the network’s depth by simply adding residual modules [39]. Common variations of ResNet, such as ResNet18, ResNet34, ResNet50, and ResNet101, are categorized based on the overall number of convolutional layers [39].

Our deep learning model, as illustrated in Figure 4a, adopts the ResNet architecture as its fundamental structure. Since our input is a 1D vector with a length of 120, we have replaced all convolutional kernels in the network with 1D convolutions. We have also modified the shortcut connections within the residual blocks to utilize 1D connections. To enhance ResNet’s ability to extract global structural features, we have introduced a multi-head attention between the global average pooling layer and the linear layer. Following the linear layer, we have removed the original activation function, as our problem is a continuous spatial nonlinear regression and the activation function will impact the continuity of the model’s output. Due to the model’s composition, which comprises 1D-ResNet and the multi-head attention mechanism, we refer to it as “1D-ResNet-MHA”.

The residual structure we employed is illustrated in Figure 4b. In each residual block, the input data on the main path goes through two sets of convolutional layers, batch normalization layers, and ReLU activation function layers. Meanwhile, the shortcut connection directly transfers input data to the second set of ReLU activation functions and after the second set of batch normalization layers. These two paths are weighted and summed to form the final output of the residual structure after passing through the second set of activation functions. During this process, when the convolutional stride on the main path is not equal to 1, it can result in a dimension mismatch between the output data of the main path and the shortcut connection. Therefore, we introduced an additional convolutional layer and batch normalization layer in the shortcut connection for projection mapping. Consequently, our residual structure exhibits two corresponding output relationships, which are as follows [39]:(1)y=F(x)+x
(2)y=F(x)+H(x)
where *x* represents the input to the residual block, *y* is the output of the residual block, F(x) denotes the output of the main path within the residual block, and H(x) represents the output of the shortcut connection when there is a dimension mismatch between the input and output.

The multi-head attention module provides an attention mechanism [40]. While convolutional layers excel at processing local information, the attention mechanism operates from a global perspective, identifying key information within the entirety of the global data while reducing or filtering out irrelevant information [40,42]. The basic principle of multi-head attention is that features are gradually grouped and processed in different heads, so that different heads can focus on extracting different types of features and re-combine those features in a meaningful way for the downstream tasks [40]. This global processing approach is particularly crucial in the context of metasurface structures, as the optical characteristics of metasurfaces depend strongly not only on local information but also on global information [43]. The workflow of the multi-head attention mechanism is depicted in Figure 4c, and its specific operations are as follows [40]: (3)$$MultiHead\left(Q,K,V\right)=Concat\left(head_{1},head_{2},...,head_{h}\right)W^{o}$$
where,
(4)$$head_{i}=Attention\left(QW_{i}^{Q},KW_{i}^{K},VW_{i}^{V}\right)=Softmax\left(\frac{QW_{i}^{Q}{\left(KW_{i}^{K}\right)}^{T}}{\sqrt{d_{i}}}\right)VW_{i}^{V}$$
where $W_{i}^{Q}$, $W_{i}^{K}$, and $W_{i}^{V}$ are projection matrices, *Q*, *K*, and *V* represent the query, key, and value, respectively, with $Q=x\cdot W_{q}$, $K=x\cdot W_{k}$, and $V=x\cdot W_{v}$, in which $W_{q}$, $W_{k}$, and $W_{v}$ are learnable matrices, and $\sqrt{d_{i}}$ is a normalization factor, in which $d_{i}$ is equal to the size of $head_{i}$.

## 5. Results and Discussion

We constructed our deep learning model using Python 3.8 and PyTorch 1.10 [44] and conducted training on an NVIDIA GeForce GTX 1080 Ti GPU [45] running Ubuntu. PyTorch is an open-source deep learning framework that integrates advantages such as ease of use, dynamic computation graphs, robust GPU acceleration support, and an active community of researchers and developers contributing to its ecosystem, and is widely used in deep learning and artificial intelligence applications [44]. After thorough experimentation and comparison of the MAE, MSE, and the Huber loss function [46], we ultimately chose the MAE as the loss function for our model. The expression for MAE is as follows: (5)$$MAE=\frac{1}{N}\sum_{i=1}^{N}\left|y_{i}-{\hat{y}}_{i}\right|$$
where *N* represents the number of samples, which is equal to 1 in this study, y represents the ground truth value, which is the coupling efficiency obtained from FDTD simulations, and $\hat{y}$ represents the predicted value, which is the coupling efficiency predicted by the neural network model. We use Adam [47] as the optimization algorithm for our model. The Adam algorithm can dynamically adjust the learning rates for each parameter during the training process, thereby enhancing model convergence and reducing reliance on the initial learning rate, and is a commonly used optimization algorithm in the field of deep learning [47].

To comprehensively assess the performance of the model, in addition to the MAE, we also use the MSE and R${}^{2}$. Their respective expressions are as follows: (6)$$MSE=\frac{1}{N}\sum_{i=1}^{N}{\left(y_{i}-{\hat{y}}_{i}\right)}^{2}$$
(7)$$R^{2}=1-\frac{\sum_{i=1}^{N}{\left(y_{i}-{\hat{y}}_{i}\right)}^{2}}{\sum_{i=1}^{N}{\left(y_{i}-{\bar{y}}_{i}\right)}^{2}}$$

All symbols in these equations hold the same meanings as defined in Equation (Equation 5), and ${\bar{y}}_{i}$ denotes the average of the ground truth values. Among these evaluation metrics, although some studies suggest that R${}^{2}$ contains more information in regression analysis than the MAE and MSE [48], the MAE provides a more intuitive representation of the difference between the predicted and actual coupling efficiencies, making it better suited for assessing whether the model’s predictive capability meets the requirements. Therefore, we primarily use the MAE as the main performance metric for the model, while also considering the performance of the MSE and R${}^{2}$. We denote the average values of MAE, MSE, and R${}^{2}$ in the similar test set as MAE${}_{s}$, MSE${}_{s}$, and R${}_{s}^{2}$, respectively, and the average values in the high-efficiency test set as MAE${}_{h}$, MSE${}_{h}$, and R${}_{h}^{2}$.

We conducted experiments to compare the model performance and training time of 1D-ResNet18, 1D-ResNet34, 1D-ResNet50, and 1D-ResNet101. We found that 1D-ResNet18 performed similarly to the other models but had a relatively shorter training time. Therefore, we selected 1D-ResNet18, with 18 convolutional layers, as the base architecture for 1D-ResNet-MHA, denoted as 1D-ResNet18-MHA.

### 5.1. Impact of Data Shift

To illustrate the impact of data shift, we trained a neural network model (1D-ResNet18) without multi-head attention on the training dataset and compared its performance on the similar test set and the high-efficiency test set. As show in Figure 5 and Table 1, the MAEs in the similar test set primarily range between 0 and 0.005, while in the high-efficiency test set, they mainly fall between 0 and 0.075. The MAE${}_{s}$ is 0.00084, while the MAE${}_{h}$ is 0.01362, which is 16 times higher. In terms of MSE and R${}^{2}$, MSE${}_{s}$ is $2.19\times 10^{-6}$ and MSE${}_{h}$ is 3.61 × 10${}^{-4}$, representing a 164-fold increase, and R${}_{s}^{2}$ is 0.972, significantly higher than R${}_{h}^{2}$, at 0.379. This indicates a significant degradation in the performance of neural networks when data shift is present compared to when data shift is absent. This decline in performance can be attributed, in part, to the neural network’s limited ability to globally process information, making it challenging to extract the true mapping relationship between the structure and coupling efficiency in the presence of data shift. Additionally, there may be an information discrepancy between the original data and the shifted data, leading to a certain degree of bias when applying the mapping rules learned from the original data to the shifted data. The impact of data shift is worthy of attention in AI-based electromagnetic solvers, since solvers often learn from randomly generated data and are then applied to predicting shifted data.

### 5.2. Addressing Data Shift and Results

Based on the analysis of data shift and its impact in the previous section, we proposed two targeted methods for mitigating the effects of data shift: mixed training and adding multi-head attention. Furthermore, we introduced a comprehensive approach that combines both methods, referred to as mixed training + multi-head attention. We conducted experiments to assess the effectiveness of these three methods in addressing data shift.

#### 5.2.1. Mixed Training Method and Comparison

The significant performance degradation of neural networks in the presence of data shift may be attributed, in part, to the information disparities between the training data and the shifted data. To enable the model to learn the information that is lacking in the training data from the shifted data, we introduced a small amount of shifted data into the training set, a technique called mixed training. Notably, this mixed training method will not become meaningless by using the existing optimized structure data in the neural network to obtain further optimized structure data. This is because the existing optimized structure data are generated using simple optimization algorithms, whereas the desired optimized structure data are unknown and may be obtained through more refined methods, making it superior to the existing optimized structure data.

To demonstrate the impact of adding different amounts of shifted data, we conducted training experiments with shifted data quantities of 0, 500, and 1000 sets using 1D-ResNet18, as depicted in Figure 6. In Figure 6a, when no shifted data are added, the MAE in the high-efficiency test set primarily falls within the range of 0 to 0.07. However, when 500 or 1000 sets of shifted data are added, the MAE primarily ranges from 0 to 0.05, indicating a narrower range compared to the former. The MAE${}_{h}$ with no added shifted data is 0.0136, with 500 sets it is 0.0102, indicating an approximately 25% degradation, and with 1000 sets, it decreases by approximately 28.7% compared to the absence of shifted data. In Figure 6b, the improvement in model performance is also evident from the MSE${}_{h}$ and R${}_{h}^{2}$, with MSE${}_{h}$ decreasing and R${}_{h}^{2}$ increasing. This demonstrates the effectiveness of the mixed training method in mitigating the impact of data shift.

#### 5.2.2. Adding Multi-Head Attention and Comparison

Another possible reason for the notable decrease in neural network performance with data shifts is the limited global information processing capacity of the neural network, which makes it challenging to accurately extract the mapping relationship from the structure data to coupling efficiency. Through experiments, the number of heads in the multi-head attention is set to eight, with each head having a size of 192. To demonstrate the effectiveness of the global information processing capability of the multi-head attention in mitigating the impact of data shift, we trained and compared 1D-ResNet18 with 1D-ResNet18-MHA (1D-ResNet18 with the multi-head attention) on the high-efficiency test set. As shown in Figure 7, the MAE distribution for 1D-ResNet18-MHA in the high-efficiency test set is slightly narrower than that of 1D-ResNet18. The MAE${}_{h}$ for 1D-ResNet18-MHA is 0.0115, which is approximately 15.4% lower than the MAE of 0.0136 for 1D-ResNet18. This shows the effectiveness of the global processing capabilities of the multi-head attention in addressing the impact of data shift.

#### 5.2.3. Mixed Training + Multi-Head Attention and Comparison

We have introduced a comprehensive approach by combining mixed training with the multi-head attention, denoted as mixed training + multi-head attention, and the corresponding neural network is called 1D-ResNet18-MHA with mixed training, where 1000 sets of shifted data are added during mixed training. We compared 1D-ResNet18-MHA with mixed training to 1D-ResNet18 in terms of the MAE distribution in the high-efficiency test set, as shown in Figure 8a. The 1D-ResNet18-MHA with mixed training exhibits a significantly narrower distribution of MAE compared to 1D-ResNet18. Specifically, the MAE${}_{h}$ for the former is 0.0087, which represents a reduction of approximately 36% compared to the latter’s MAE${}_{h}$ of 0.0136. We also compared the mixed training + multi-head attentionwith the previous mixed trainingand adding multi-headmethods. As shown in Figure 8b and Table 2, the 1D-ResNet18-MHA with mixed training outperforms the 1D-ResNet18 with mixed training and 1D-ResNet18-MHA in terms of MAE${}_{h}$, MSE${}_{h}$, and R${}_{h}^{2}$. This strongly demonstrates the effectiveness of our mixed training + multi-head attention in addressing data shift.

## 6. Summary

A subtle yet often overlooked issue in utilizing a deep learning electromagnetic solver to design nano-structured metamaterials as electromagnetic solvers is the problem of data shift. This refers to the situation where neural networks are trained on a training dataset generated with structure data in a random manner but are eventually applied to predict selectively optimized structure data during the optimization process. This paper, for the first time in the field of AI-based electromagnetic solvers, illustrates the presence and impact of the data shift issue and provides potential solutions. Taking a one-dimensional grating coupler as an example, we have employed a random generation method to create a training set and a similar test set with low efficiency, with structure data optimized using a genetic algorithm serving as the high-efficiency test set, which emulates the nano-structured metamaterial data that deep learning models need to predict during the design process using optimization algorithms. We have analyzed the probability density of labels in the high-efficiency test set and training set, along with the principal components of the input data, to demonstrate the presence of data shift. Adopting 1D-ResNet-MHA as the primary framework for our deep learning model, we have demonstrated the detrimental impact of data shift on the neural network’s performance by examining the performance of 1D-ResNet18 without the multi-head attention on both the similar test set and the high-efficiency test set. Based on our analysis for the causes of data shift, we propose three methods to mitigate its effects, namely, mixed training, adding multi-head attention, and a comprehensive approach that combines both methods. The experimental results indicate the effectiveness of our methods in addressing data shift. Particularly, the combination of mixed training and multi-head attention significantly reduces the MAE, by approximately 36%, when applied to data affected by data shift. Our approach holds significant promise in the domain of AI-based electromagnetic solvers for nano-structure metamaterials, as it addresses the challenge of deep learning models learning from random data but being applied to predicted shifted data during the optimization process.

## Figures and Tables

**Figure 1 nanomaterials-13-02778-f001:**
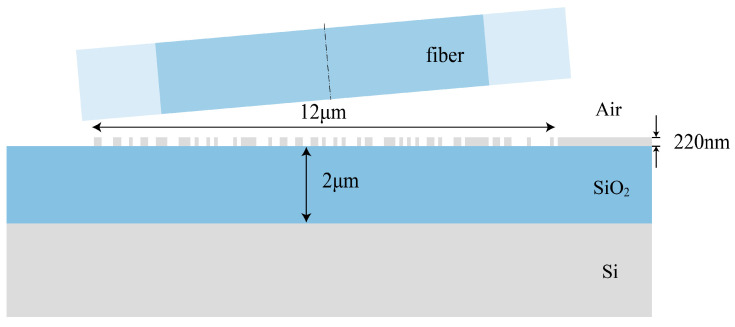
Two-dimensional cross-sectional schematic of a one-dimensional non-uniform grating coupler.

**Figure 2 nanomaterials-13-02778-f002:**
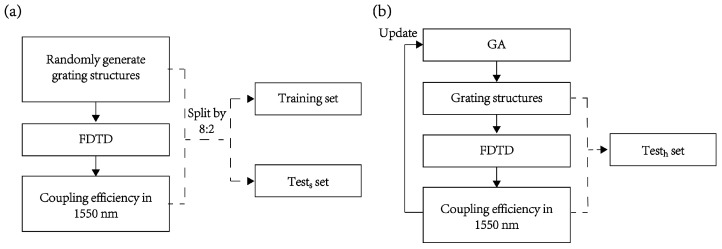
Dataset generation workflow. (**a**) Process for generating the training set and the similar test set; (**b**) process for generating the high-efficiency test set.

**Figure 3 nanomaterials-13-02778-f003:**
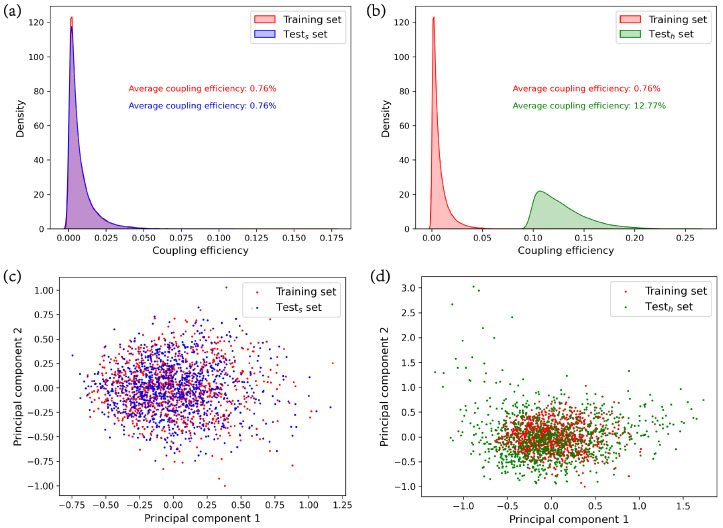
Comparison of data shift among datasets. (**a**) Probability density plots of labels in the training set and the similar test set; (**b**) probability density plots of labels in the training set and the high-efficiency test set; (**c**) distribution of two principal components of inputs in the training set and the similar test set; (**d**) distribution of two principal components of inputs in the training set and the high-efficiency test set.

**Figure 4 nanomaterials-13-02778-f004:**
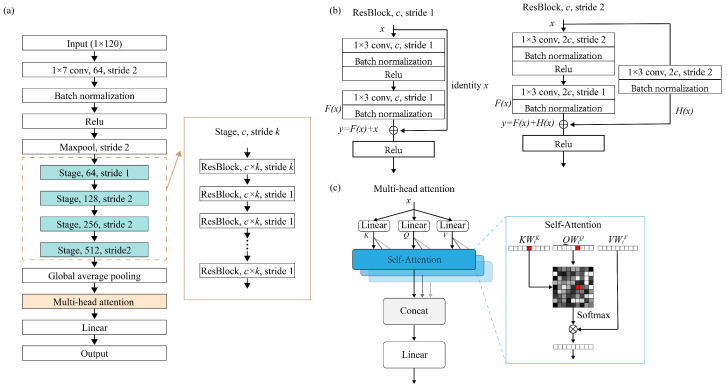
1D-ResNet-MHA. (**a**) Basic framework of 1D-ResNet-MHA; (**b**) residual block; (**c**) multi-head attention mechanism.

**Figure 5 nanomaterials-13-02778-f005:**
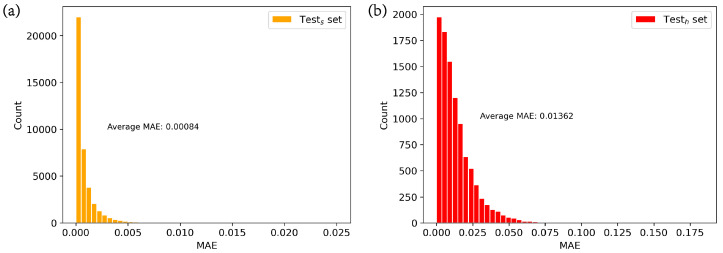
Impact of data shift. (**a**) MAE distribution in the similar test set; (**b**) MAE distribution in the high-efficiency test set.

**Figure 6 nanomaterials-13-02778-f006:**
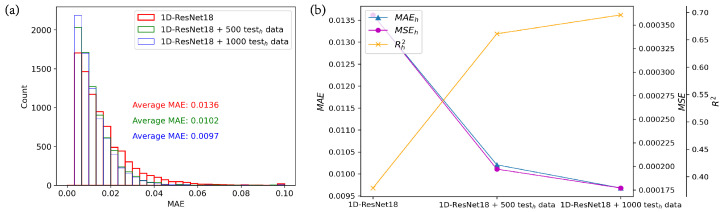
Comparison of adding different amounts of shifted data. (**a**) Comparison of MAE distribution in the high-efficiency test set; (**b**) comparison of MAE, MSE, and R${}^{2}$ on the high-efficiency test set.

**Figure 7 nanomaterials-13-02778-f007:**
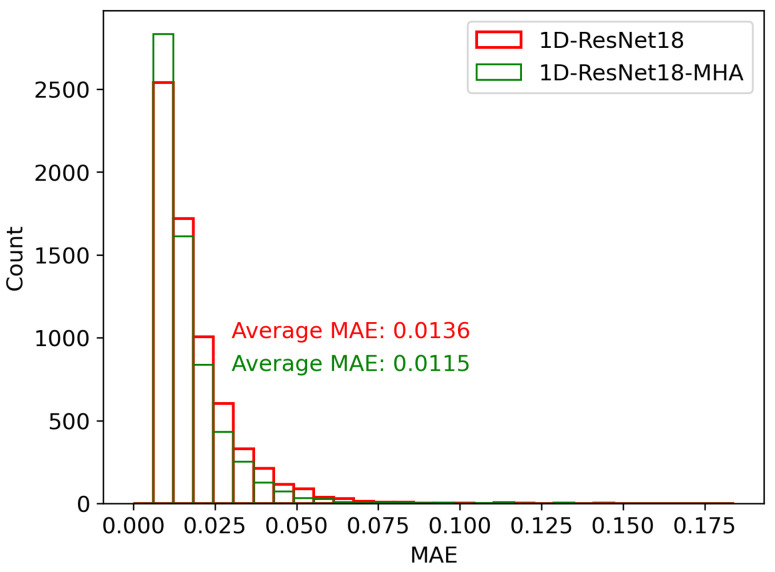
Comparison of MAE distribution between 1D-ResNet18 and 1D-ResNet18-MHA in the high-efficiency test set.

**Figure 8 nanomaterials-13-02778-f008:**
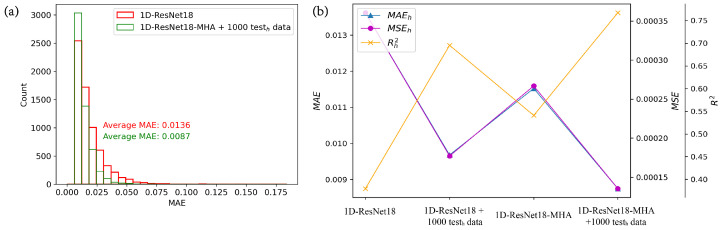
Comparison of neural network performance on high-efficiency test set. (**a**) Comparison of MAE distribution between 1D-ResNet18 and 1D-ResNet18-MHA with mixed training in the high-efficiency test set; (**b**) comparison of models using different methods on the high-efficiency test set.

**Table 1 nanomaterials-13-02778-t001:** Comparison of MAE, MSE, and R${}^{2}$ on the similar test set and the high-efficiency test set.

Test Set Name	MAE	MSE	R${}^{2}$
Test${}_{s}$ set	0.00084	2.19 × 10${}^{-6}$	0.972
Test${}_{h}$ set	0.0136	3.61 × 10${}^{-4}$	0.379

**Table 2 nanomaterials-13-02778-t002:** Comparison of models using different methods on the high-efficiency test set.

Method	MAE	MSE	R${}^{2}$
1D-ResNet18	0.0136	3.61 × 10${}^{-4}$	0.379
1D-ResNet18 + 1000 Test${}_{h}$ data	0.0097	1.77 × 10${}^{-4}$	0.695
1D-ResNet18-MHA	0.0115	2.67 × 10${}^{-4}$	0.541
1D-ResNet18-MHA + 1000 Test${}_{h}$ data	0.0087	1.35 × 10${}^{-4}$	0.767

## Data Availability

The data that support the results are not published at this time, but may be obtained from the authors upon reasonable request.

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
