# Peer review of "Utilizing Mixed Training and Multi-Head Attention to Address Data Shift in AI-Based Electromagnetic Solvers for Nano-Structured Metamaterials"

_nanomaterials, 2023, doi:10.3390/nano13202778_

Round 1
Reviewer 1 Report
1. Is it possible to be more specific on the resources regarding the computational packages?
2. Have you tested any more complicated structures regarding the dimensions?
The language and the grammar of the paper are in very good level.
Author Response
Dear Reviewer:
Thank you very much for taking the time to review our manuscript entitled “Utilizing Mixed Training and Multi-Head Attention to Address Data Shift in AI-Based Electromagnetic Solvers for Nano-Structured Metamaterials”. Your constructive and insightful comments have been tremendously valuable in improving our work. In this revision, we have addressed all of these suggestions. Except the advice mentioned in the comments, we identified find some additional inappropriate points and made corresponding revisions in the manuscript. All the revisions are highlighted in red color. Our point-to-point responses to the comments are listed in the attached document.
Due to the website's limitation of allowing only one file to be attached, we are unable to include the revised manuscript for your review. However, if you have any specific requests, we would be more than willing to send it to you via email.
With best wishes.
Yours sincerely,
Prof. Sailing He, Fellow of IEEE, OSA, SPIE, and the Electromagnetics Academy Director, National Engineering Research Center for Optical Instruments
Zhejiang University, Hangzhou, 310058, China.
Email: sailing@zju.edu.cn

Reviewer 2 Report
Zeng et al. have investigated the impact and solutions of the data shift in AI based electromagnetic solvers for nano structure metamaterials. They have used one-dimensional grating coupler as an example to compare the coupling efficiency distribution of randomly generated device structures. The results obtained by the author and impact of the work is very objective and a through investigation has been done. My comments are listed below
1. A detailed description of Lumerical FDTD set up should be described in the paper, how did the author calculated the coupling efficiency from Lumerical FDTD. A detail explanation with possible scheme is needed.
Minor edits and grammar change needed.
Author Response

(The authors gave the same response as above.)

Reviewer 3 Report
Dear Authors,
Please find attached the PDF file with comments as sticky notes.
Kind regards.

Minor editing of English language required.
Author Response

(The authors gave the same response as above.)
